# *Lawsonia intracellularis* regulates nuclear factor-κB signalling pathway during infection

**Huan W. Yang**[1], **Tuanjun Hu**[2], **Tahar Ait-Ali**[3]*

**1** Department of Biochemistry, The University of Illinois Champaign-Urbana, Champaign, IL, United States of America, **2** The Roslin Institute and Royal (Dick) School of Veterinary Studies, University of Edinburgh, Midlothian, United Kingdom, **3** ANSES Fougères Laboratory, Fougères, France

* tahar.aitali@anses.fr

**Data Availability Statement:** the data within the manuscript and Supporting Information files are available: https://doi.org/10.5281/zenodo.12705803.

## Abstract

*Lawsonia intracellularis* is the etiological agent of proliferative enteropathy (PE) in pigs, horses and wide range of mammals. Little is known about the role of innate immune response during *L. intracellularis* infection. In this study, we investigated the nuclear factor-κB (NF-κB)-regulated immune response against infection of a clinical strain Dkp23 and a live-attenuated Enterisol vaccine strain in PK-15 cells. We found that expression of NF-κB target genes *TNF-α*, *IFN-γ*, *IL-6* and *IL-8* were modulated during the course of infection. At 5 dpi, there was a significant increase in p65 NF-κB activation, including protein nuclear translocation and phosphorylation, synchronous with the induction of *IL-6*, *IFN-γ* and *IL-8* expression in *L. intracellularis* infected cells, especially for Enterisol vaccine strain-infected cells. This result suggests that NF-κB signalling level is induced when *L. intracellularis* bacterial load peaks at 5 dpi. The induction of pro-inflammatory cytokines expression is consistent with the decreased viability of *L. intracellularis*-infected cells especially that of the vaccine strain. There were no significant changes in NF-κB signalling between vaccine and Dkp23 infection in PK-15 cells, except for moderate levels of differences in NF-κB target genes expression which might be a reflection of differences in intracellular bacterial load. Overall, the data presented here indicate a correlation between the induction of NF-κB signalling and the *L. intracellularis* bacterial load in PK-15 cells.

## Introduction

*Lawsonia intracellularis* is a Gram-negative obligate intracellular bacteria and the causative agent of proliferative enteropathy (PE) in pigs, horses and a wide range of mammalian species [1–5]. *L. intracellularis* infected ileums harbour rod shaped bacteria clustered in the apical cytosol of the intestinal crypt cells [6,7]. The main hallmark of *L. intracellularis* infection is the macroscopically thickening of intestinal mucosal lining due to hyperproliferation of epithelial cells [1,3,8–10]. As an obligate intracellular bacterium, little is known how the bacteria survives and replicates within the host cells, and how such interaction facilitate the disease pathogenesis of PE. Furthermore, the recent observation that *L. intracellularis* is able to internalize and survive within phagolysosomal environment of pig macrophages points out toward that this

**Funding:** The author(s) received no specific funding for this work.

**Competing interests:** The authors have declared that no competing interests exist.

pathogen is able to build sophisticated mechanisms to proliferate intracellularly [11,12]. Limited studies of the molecular pathogenesis of PE indicate that intestinal epithelial cell proliferation is associated with the down-regulation of specific host systems involved in nutrients and solute absorption, as well as the maintenance of mucosal integrity and the disruption of autophagy [12–16]. Another consistent molecular pathogenesis marker associated with PE is the reduction in goblet cells secretory products such as Mucin-2, Resistin-like molecule β and Trefoil factor 3 [13,14,16]. Thus, the loss of goblet cells secretion especially that of Mucin-2, combined to the disruption of autophagy might aggravate the clinical manifestation of PE by increasing the risk of secondary infections and altering the intestinal epithelium homeostasis.

Innate immune responses represent the first line of defence against invading pathogens. Upon recognition of pathogen specific patterns, various signalling pathways are activated to provide cell and humoral-mediated responses in aiding the removal of infectious agents [17,18]. One such pathway is the NF-κB signalling pathway. This pathway comprises the canonical and alternative pathways, with the former involved in innate, inflammatory, and adaptive immune responses, while the latter is involved in the regulation of lymphoid organ development, B-cell function, and adaptive immunity. NF-κB family members consists of five subunits, which are p65 (RelA), RelB, cRel, p50 and p52 [19–21]. Given the critical role that NF-κB plays in cell death, inflammation, and immune response, many pathogen effector proteins target the NF-κB pathway. Since intracellular pathogens rely heavily on the host cell mechanisms to grow and replicate, multiple pathogenic and virulence factors are known to attenuate NF-κB signalling pathway to ensure the survival of both the pathogen as well as the host cell. Previous studies have identified multiple host factors involved at every stage of NF-κB signalling as targets of intracellular pathogens-derived factors and the list is expected to grow, with more host-pathogen interaction studies being carried out [20,21]. The role of NF-κB signalling in disease pathogenesis and clearance of *L. intracellularis* infection is largely unexplored. Although the upregulation of interferon responsive genes was reported in *L. intracellularis*-infected IEC-18 48 hours post infection (hPI) [22], it is speculative whether NF-κB signalling is induced for a longer period of time during infection as it could acts as both a target of interferon stimulated pathway and an inducer of interferons. Smith and colleagues [23] showed that knockout of IFN-γ receptor in mice delayed the clearance of *L. intracellularis* infection and increased mortality of infected mice, suggesting an important role of IFN-γ mediated immunity in clearing *L. intracellularis* infection. However, IFN-γ cytokine is known to be of very low to undetectable level in sera of pigs diagnosed of non-haemorrhagic PE and downregulated level in *L. intracellularis* infected foals at the peak of infection [24–27]. It is unclear whether IFN-γ mediated immune responses play a role at early stages of *L. intracellularis* infection by restricting bacterial growth, hence its inhibition leads to increase other bacterial load and susceptibility to secondary infections. Previous studies have indicated that immunosuppressive mechanisms may be operating during *L. intracellularis* infection [24,28–31]. Whether the changes observed at the peak of *L. intracellularis* infection are associated with an alteration of innate immunity within infected cells such as NF-κB signalling remains ill defined.

In this paper, the level of NF-κB signalling factors was measured in epithelial cells challenged with either a clinical isolate Dkp23 or a live-attenuated vaccine strain in cell culture [32]. In conclusion, this study provides initial evidences that, NF-κB signalling is induced during infection by *L. Intracellularis* within the host cells.

## Materials and methods

### *L. intracellularis* strains

Two strains of *L. intracellularis* were used for this work: a) the clinical strain Dkp23 and b) the live-attenuated vaccine strain (Enterisol, Boehringer Ingelheim, Burlington, ON). The Dkp23 strain originated from pigs diagnosed with heavy haemorrhagic form of PE. This strain was propagated in Mouse Intestinal Epithelial Cells (IEC-18) for 23 passages (Courtesy of Pr. David GE Smith and Eleanor Watson, the Moredun Research institute). The live-attenuated vaccine was provided as a lyophilised pellet which was dissolved in PBS or sterile water, yielding approximately $5.3 \times 10^5$ organisms per mL (Enterisol Ileitis®, Boehringer Ingelheim Vetmedica GmbH). The genome sequence of both strains have been determined previously [32].

### Cell culture and bacterial propagation

For cell culture and bacterial propagation the Porcine Kidney 15 (PK-15) cells (PK-15, ATCC® CCL33™) was used. PK-15 cells were grown in Dulbecco's Modified Eagle Medium (DMEM, Thermo Fisher Scientific, UK) supplemented with 10% of heat inactivated Fetal Bovine Serum (Thermo Fisher Scientific, UK), 1% GlutaMAX™ (Thermo Fisher Scientific, UK) and 100 units ml$^{-1}$ of penicillin/streptomycin (Thermo Fisher Scientific, UK) in a humidified chamber with an atmosphere of 5% carbon dioxide and 95% air at 37˚C. Cells were passaged every 3–4 days at a ratio of 1:4 using TrypLE-EDTA (0.25%) (Thermo Fisher Scientific, UK). Antibiotics supplement was omitted for at least one passage number before being used for infection.

Infection was carried out in freshly trypsinized cells, seeded at 30% confluency in Nunc™ 6-wells or 24-wells culture dishes (Thermo Fisher Scientific, UK) in their respective medium without penicillin/streptomycin supplement. Cells were then co-cultured with *L. intracellularis* at a multiplicity of infection (MOI) of 1. To increase the rate of infection, cell culture dishes were centrifuged at 2,020 *g* for 10 min as described in previous study [33]. Cell culture dishes were then placed into an anaerobic jar (Don Whitley Scientific, York) filled with an anaerobic tri-gas mixture consisting of 80% nitrogen, 10% hydrogen and 10% carbon dioxide, as described in previous study [34]. The temperature of anaerobic jar was maintained at 37˚C by placing it into an incubator and the cell culture medium was changed every 2–3 days.

Lawsonia bacterial inoculant is prepared from infected cells at 5 days post infection (dpi). Cells were collected using TPP 240 mm cell scraper (Scientific Laboratory Supplies, UK) and passed through a 1½ inch, 21 Gauge hypodermic needle (Sigma Aldrich) for 10–20 times. Cell debris were spun down at 100 *g* for 5 min while the supernatant containing bacteria was added onto fresh cells. For freezing of *L. intracellularis* aliquots, Dimethyl sulfoxide (D2438 Sigma Aldrich) was added to a final volume of 10% and aliquots were stored at -80˚C.

### Poly (I:C) treatment of PK-15 cells

As a positive control, 10 μg/ml of Polyinosinic-polycytidylic acid (Poly (I:C), Invitrogen) was used to treat PK-15 cells for activating NF-κB [35,36], the treated cell were cultured for 48 h, in the same conditions as described previously for *L. intracellularis* infection.

### Immunofluorescence staining of *L. intracellularis* and p65-NF-κB in PK-15 cells

PK-15 cells were cultured on 13 mm round glass coverslips (Thermo Fisher Scientific) in 24-well culture dishes and infected with *L. intracellularis* using the conditions stated previously. For immunofluorescence imaging, supernatant was removed and cells were fixed with

4% paraformaldehyde for 20 min at room temperature. Cells were permeabilised with 1% Triton-X in Phosphate Buffered Saline (PBS) for 10 min and blocked with 5% Bovine Serum Albumin (BSA) in PBS for 30 min at room temperature. Primary antibody incubation was carried out for 1 hour at room temperature using monoclonal VPM53 antibody detecting *L. intracellularis* (Mouse, 1:400 dilution, University of Edinburgh) and anti-p65 antibodies (Rabbit, 1:400 dilution, #8242 Cell Signalling) diluted in PBS containing 2% BSA. Slides were washed three times with PBS at room temperature, followed by 1hour incubation with secondary antibodies Alexa-Fluor-488 goat anti-mouse (1:1000, Thermo Fisher Scientific) and Alexa-Fluor-647 goat anti-rabbit (1:1000, Thermo Fisher Scientific) diluted in PBS containing 2% BSA. Cells were washed three times with PBS before incubation with 6-diamidino-2-phenylindone (DAPI, 1:500 in water, Thermo Fisher Scientific) at room temperature for 10 min. Cells were washed three times in PBS and the coverslips were mounted onto Superfrost™ Plus Microscope (Thermo Fisher Scientific) using Vision PermaFluor aqueous mounting medium (Thermo Fisher Scientific). Slides were observed using LSM700 confocal laser scanning microscope (Carl Zeiss). Images were captured and processed using Zen Blue software (Carl Zeiss). To quantify the p65 nuclear staining

P65-NF-κB nuclear staining of uninfected and *L. intracellularis* infected PK-15 cells were quantified using ImageJ1.49S (ImageJ, U. S. National Institutes of Health). For each time points p65-NF-κB staining (fluorescence) intensity of nuclei was measured in at least 4 random fields. Mean fluorescence readings were acquired. Nuclear p65-NF-κB staining intensity of *L. intracellularis* infected PK-15 cells was normalised to that of uninfected controls at each time points. Thus, data present a ratio of nuclear p65-NF-κB staining of *L. intracellularis* infected PK-15 cells to that of uninfected controls.

## QPCR analysis of *L. intracellularis* genomic copy number

DNA extraction was performed on uninfected or *L. intracellularis*infected PK-15 cells using DNeasy Blood and Tissue Kit (Qiagen) using the manufacturer's protocol. Briefly, cells were collected using TPP 240 mm cell scraper (TIS7040, Scientific Laboratory Supplies, UK). Cells were pelleted at 200 *g* for 5 min and treated with a DNeasy Blood and Tissue Kit (Qiagen). The quantity and quality of extracted DNA was assessed using a Nanodrop ND1000 spectrophotometer (NanoDrop Technologies Inc., Wilmington, DE, USA). Primers for qPCR were designed against aspartate ammonia lyase gene (Asp, for primer sequence, refer to S1 Table) of *L. intracellularis* as described in previous studies [32]. QPCR was carried out using Brilliant III Ultra-Fast SYBR® Green QPCR Master Mix (Agilent Technologies) following manufacturer's protocol. Reactions were performed using LightCycler 480 (Roche).

RNA preparation from PK-15 cells was performed on cell pellet using RNeasy Mini Kit (Qiagen). Briefly, cells were collected using TPP 240 mm cell scraper (Scientific Laboratory Supplies, UK). Cells were pelleted at 200 *g* for 5 min and processed with the manufacturer's protocol of RNeasy Mini Kit (Qiagen). RNA samples of uninfected and *L. intracellularis* infected pig ileums were from a previous study [16]. Briefly, Trizol (Thermo Fisher Scientific) was used to extract total RNA. Total RNA was further purified using RNeasy Mini Kit (Qiagen). RNase-Free DNase Set (Qiagen) was used to remove contaminating DNA from the purified RNA samples. The quantity and quality of RNA samples were assessed using a Nanodrop ND1000 spectrophotometer (NanoDrop Technologies Inc., Wilmington, DE, USA) and Agilent 2100 bioanalyser (Agilent Technologies) respectively.

## Real time quantitative PCR (RT-qPCR)

RT-qPCR was carried out using RNA extracted from uninfected and *L. intracellularis* infected PK-15 cells at 2 and 8 hpi and 1, 3 and 5 dpi, as well as from uninfected. OligoPerfect™

Designer (Thermo Fisher Scientific) was used to design primers targeting the region between exons pair flanking an intron (S1 Table). Brilliant III Ultra-Fast SYBR Green RTqPCR Master Mix kit (Agilent Technologies) was used following manufacturer's protocol. RT-qPCR reactions were performed using Stratagene MX3000P (Stratagene), using the maunfacturer's recommended thermal profile: 10 min at 50˚C; 3 min at 95˚C and 40 cycles of 95˚C 20 s, 60˚C 20 s. Each RNA sample is run as triplicate. RT-qPCR results were normalised to that of housekeeping gene, GAPDH mRNA level and to uninfected samples.

## Cell viability assay

The viability of uninfected and *L. intracellularis* infected PK-15 cells at 1, 3 and 5 dpi were assessed using CellTiter 96® AQ$_{ueous}$ Non-Radioactive Cell Proliferation Assay (MTS) (Promega UK) following the manufacturer's protocol. Briefly, 20 μL (per 100 μL of cell culture) of the reconstituted MTS-PMS solution was added directly into the cell culture medium and incubated in a humidified incubator with an atmosphere of 5% carbon dioxide and 95% air for 2 h. MTS is reduced by viable cells to produce soluble formazan products with peak absorbance at 490nm. Absorbance of the formazan products at 490nm was measured using Cytation™3 microplate reader (BioTek). The amount of formazan products present, represented by the absorbance reading at 490 nm is directly proportional to the amount of viable cells. Corrected absorbance value was measured as follows: corrected absorbance reading = absorbance value of wells containing uninfected or infected cells–absorbance value of wells containing cell culture medium only. Percentage viability of *L. intracellularis* infected PK-15 cells were measured by comparing its absorbance value at 490 nm to that of uninfected cells.

## Analysis of IL-8 cytokine levels using cytokine array

Culture medium were collected at 1, 3, and 5 dpi with either strain DKp23 or Vaccine or mock-infected. The levels of IL-8 cytokine were determined in triplicate using the porcine cytokine antibody array A (# ab197479, Abcam) according to the manufacturer's instructions.

## Western blot analysis

Western blot analysis were run as previously described [37]. Phosphorylated-NF-κB-p65 and p65-NF-κB were detected with a rabbit anti-Phospho-NF-κB p65 antibody (Ser536, #3033, Cell Signaling Technologies) and a rabbit anti NF-κB p65 antibody (1/1000, #8242 Cell Signaling Technologies), respectively. For the detection an anti-rabbit HRP-conjugated antibody (#7074, Cell Signaling Technologies) and an ECLTM Western Blotting Detection Reagents system (GE Healthcare Life Sciences, Little Chalfont, UK) were used according to manufacturer' instructions.

## Statistical analysis

Microsoft Excel was used to generate the graphs (Microsoft, Redmond, WA, USA). Student's two-tailed t-test with assumed unequal variance was used to determine statistical significance in the observed differences. Experimental readings such as those from RT-qPCR, fluorescence intensity, cell viability etc. were compared between uninfected and infected samples, unless stated otherwise. All data is shown as mean value ± standard deviation. A p-value lower than 0.05 is considered as statistically significance. Five points stars (★) on graphs represents *p*-values for differences with statistical significance, when compared to uninfected samples. ★ denotes *p<0.05*, ★★ denotes *p<0.005*, ★★★ denotes *p<0.0005*, ★★★★ denotes *p<0.00005*.

## Results

### Growth of *L. intracellularis* Dkp23 and vaccine strain in epithelial cells

PK-15 cells were infected at a MOI of 1 with the pathogenic *L. intracellularis* strain Dkp23 and a live-attenuated vaccine strain and the bacterial growth was assessed daily until 5 dpi by real-time qPCR using bacterial genome specific PCR primers as described in S1 Table. As expected both strains had similar amount of bacterial DNA at 1 dpi (p>0.05), while the vaccine strain exhibited a higher growth rate than that of Dkp23 at 3 and 5 dpi (Fig 1A), Immunofluorescence staining of *L. intracellularis* antigen indicated an increase in bacterial antigen staining at 3 and 5 dpi therefore verifying that PK-15 cells could support the growth of both Dkp23 and vaccine strains of *L. intracellularis* (Figs 1B and S1).

### Reduced host cell viability during *L. intracellularis* infection

Cell viability assay was measured during the course of the infection of PK-15 cells by the *L. intracellularis* strains Dkp23 and the vaccine at 1, 3 and 5 dpi (Figs 2 and S2). Both Dkp23 and vaccine strain-infected cells showed significant reduction of cell viability 5 dpi (Dkp23, 73.8 ±1.0; vaccine, 57.8 ± 6.5; *p<0.005*).

### Modulation of cytokine expression during *L. intracellularis* infection

To determine the regulation of cytokine during the infection by Dkp23 or the vaccine strain, levels of *TNFα*, *IFN-γ*, *IL-8* and *IL-6* transcripts were monitored (Fig 3). There were no significant differences (*p>0.05*) in *TNF-α*, *IFN-γ*, and *IL-8* mRNA levels for both strains 2 and 8 hpi as compared to uninfected control (Fig 3A, 3B and 3D). *IL-6* mRNA level of vaccine strain infected cells was significantly reduced (*p<0.05*) at 2 hpi (Fig 3C). At 1 and 3 dpi, *IFN-γ*mRNA level was significantly reduced with the vaccine strain infection only (*p<0.05*) (Fig 3A). At 5 dpi, *IL-6* mRNA was significantly induced (*p<0.05*) with both strains (Fig 3C). Interestingly, *IFN-γ* mRNA level was induced in vaccine strain infected cells 5 dpi (*p<0.005*) (Fig 3A). *TNF-α* mRNA level remained transiently reduced in both infected cells at 1 and 3 dpi (*p<0.05*) (Fig 3B). *IL-8* mRNA levels were significantly induced at 3 dpi (*p<0.005*) and at 5 dpi (*P<0.0005*) in both Dkp23 and vaccine strain infected cells as compared to uninfected controls (Fig 4C). Consistent with this result IL-8 cytokine level showed consistent induction at 3 and 5 dpi above the levels of uninfected controls regardless of the strain of L. *intracellularis* used (S3 Fig). As control experiment *IFN- γ*, *TNF-α*, *IL-8* and *IL-6* transcript levels were also measured in PK15 cells treated with poly (I:C) for 2, 8, 24 and 48 h. *IFN- γ* mRNA level was significantly induced at 48 h post treatment (hpt) (*p<0.005*) while both *TNF-α* and *IL-6 mRNA* levels were significantly induced (*p<0.05*) at 8 and 24 hpt, as compared to untreated controls (S4 Fig). *IL-8* mRNA level in Poly (I:C) treated cells remained significantly induced (*p<0.05*) at all-time points. Taken together, these results suggest that cytokine production is modulated during infection by *L. intracellularis* regardless of the strain pathogenicity.

### Both strains Dkp23 and vaccine promote nuclear translocation and the phosphorylation of p65 NF-κB 5 dpi

The cellular localisation of p65 NF-κB was monitored at 2 and 8 hpi and 1, 3 and 5 dpi (Figs 4A and S5). We found that at 5 dpi, there was a significant increase in p65 NF-κB nuclear staining regardless of the bacterial strain used (*p<0.0005*) (Fig 4B). Consistent with this nuclear translocation, we found a strong accumulation of phospho-p65 NF-κB, 5 dpi with both *L. intracellularis* strains (Fig 4C). In contrast, Strong nuclear translocation of p65 NF-κB

A

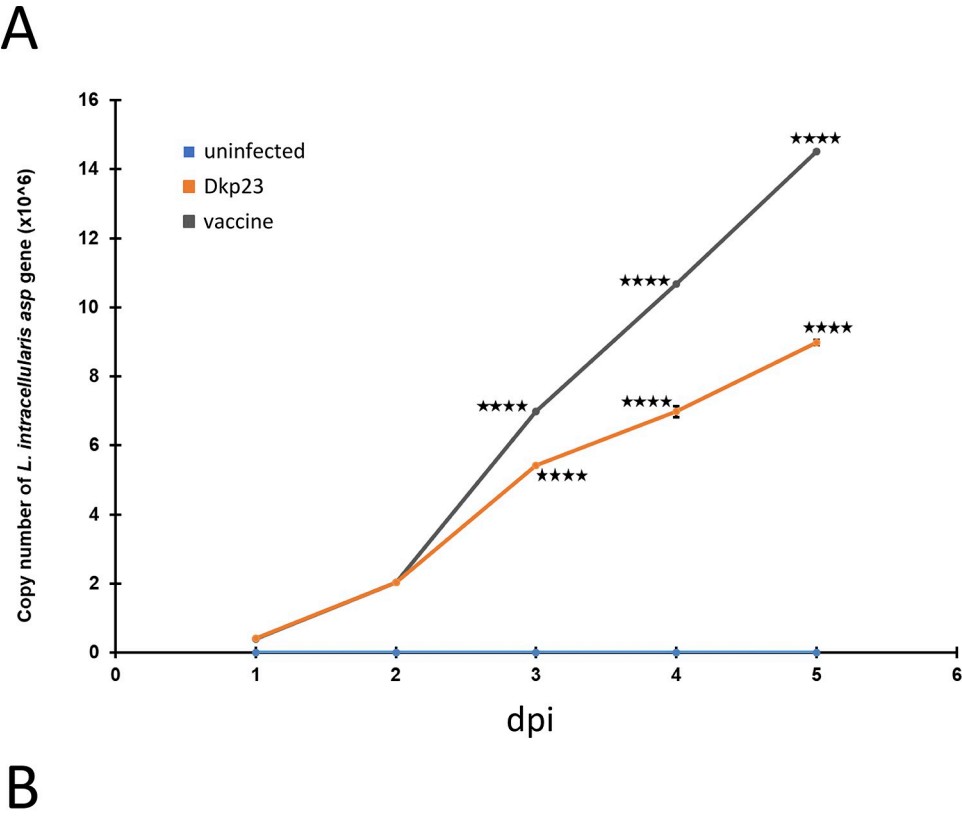

B

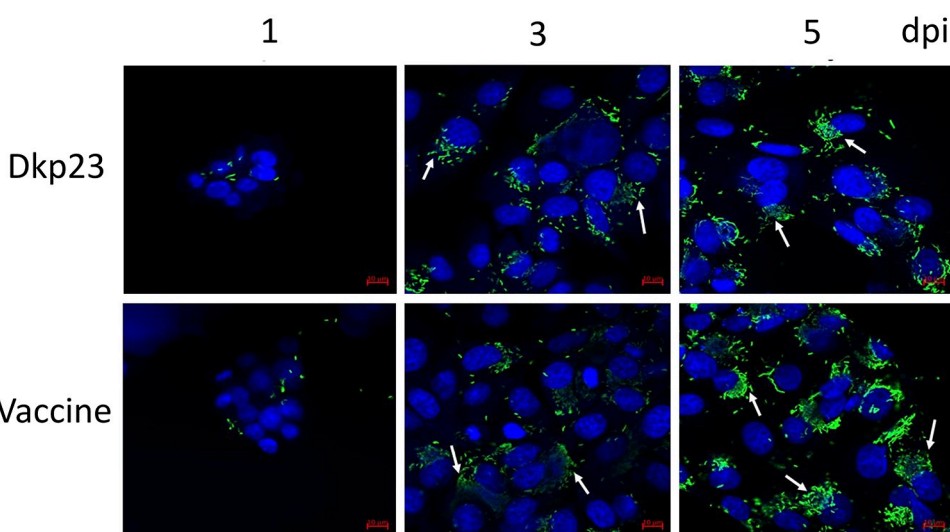

**Fig 1. Growth of *L. intracellularis* strains Dkp23 and Vaccine in PK-15 cell lines.** A qPCR analysis of *L. intracellularis* genomic copy number in Dkp23 (orange), Vaccine-infected (grey) and uninfected (blue) PK-15 cells at 1, 2, 3, 4 and 5 dpi. Mean values ± standard deviations are shown. Y-axis represents the copy number of *L. intracellularis* genome. **B** Immunofluorescence staining of rod-shaped *L. intracellularis* strains Dkp23 and Vaccine using mouse monoclonal antibody, VPM53, in uninfected and infected PK-15 cells at 1, 3 and 5 dpi, followed by Alexa-488 conjugated secondary antibody (Green). Nuclei were counterstained with DAPI (Blue). White arrows pointed at the infected cells with more than 30 bacteria in cluster. Scale Bar: 10 µm.

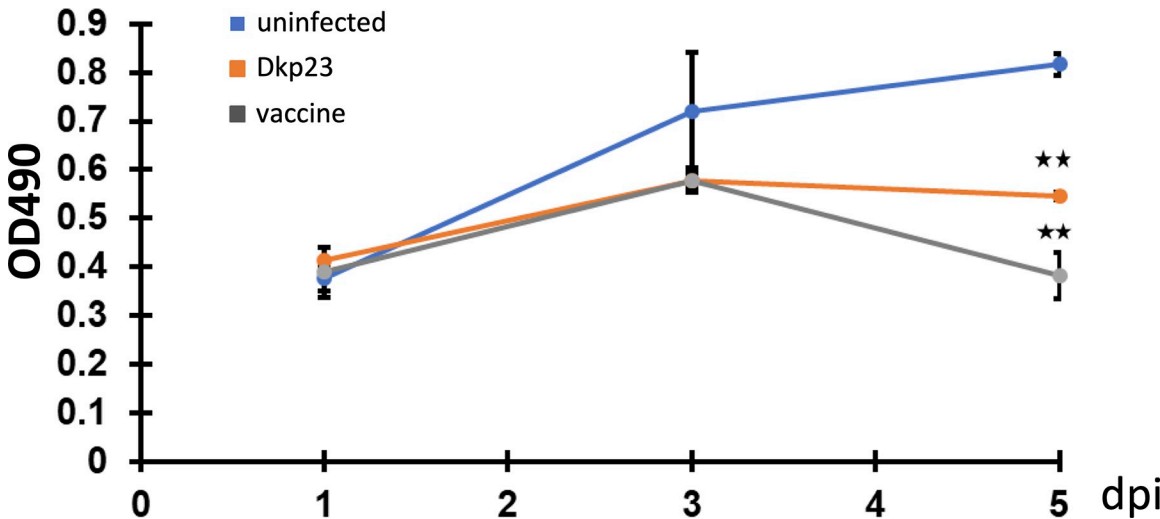

**Fig 2. Reduced cell viability in Dkp23- and Vaccine-infected PK-15 cells.** Corrected absorbance value of formazan product at 490 nm in uninfected, Dkp23 and Vaccine infected PK-15 cells at 1, 3 and 5 dpi. Mean values ± standard deviations are shown. Y-axis represents the corrected absorbance value at 490 nm.

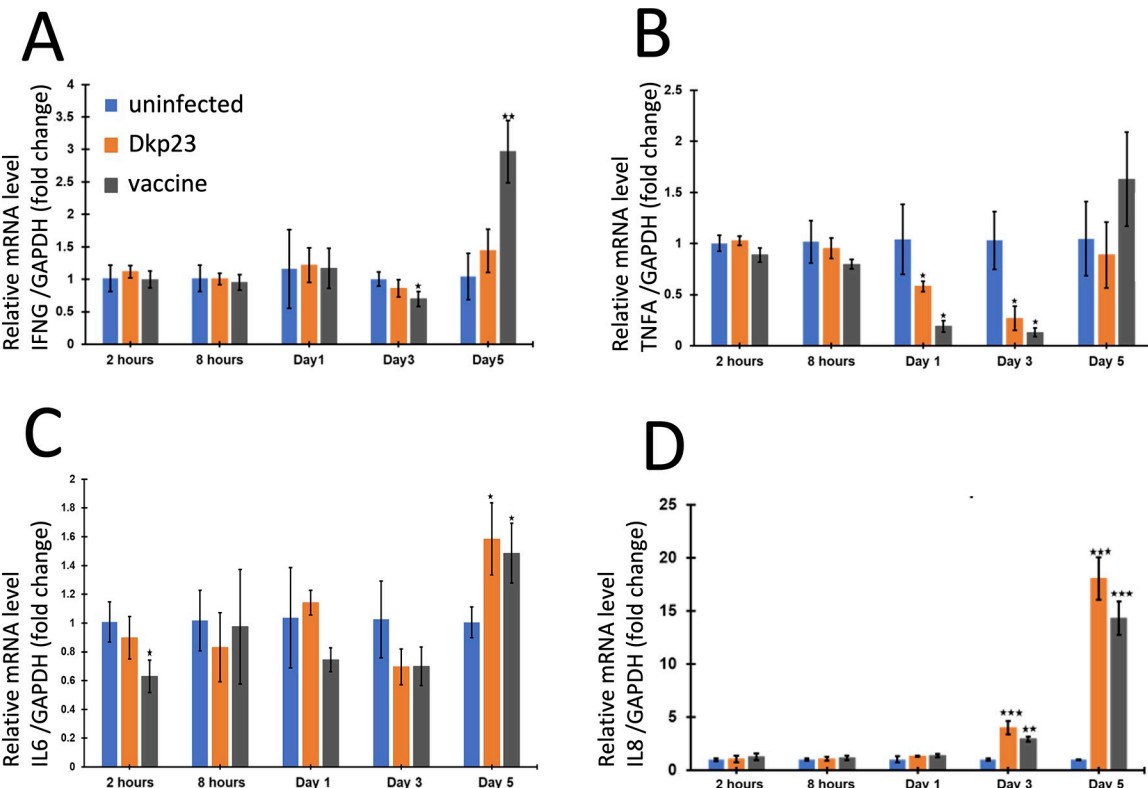

**Fig 3. Regulation of NF-κB target genes mRNA levels in *L. intracellularis* infected PK-15 cells. Transcript levels for A)** *IFN-γ*, **B)** *TNF-α*, **C)** *IL-6* and **D)** *IL-8* mRNA transcript levels were measured by RT-qPCR at 2 and 8 hpi and 1 3 and 5 dpi with Dkp23 or vaccine strains and data were normalised to the level GAPDH transcript of untreated cells. Mean values ± standard deviations are shown. Y-axis represents fold change in mRNA transcript levels as compared to that of untreated controls.

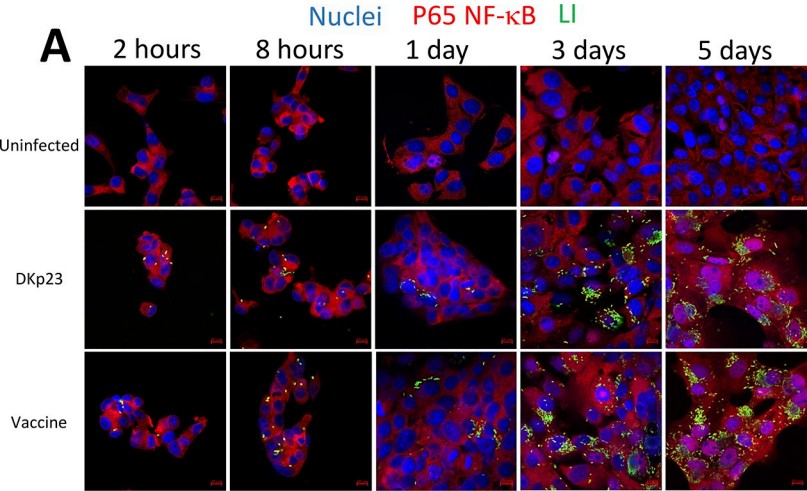

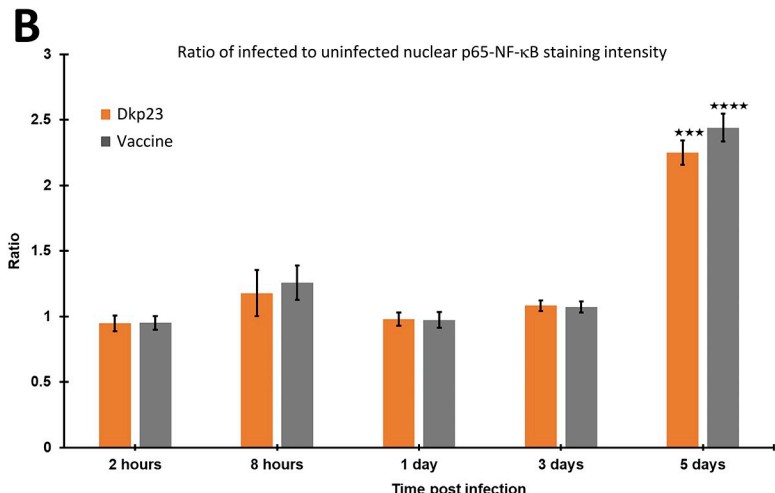

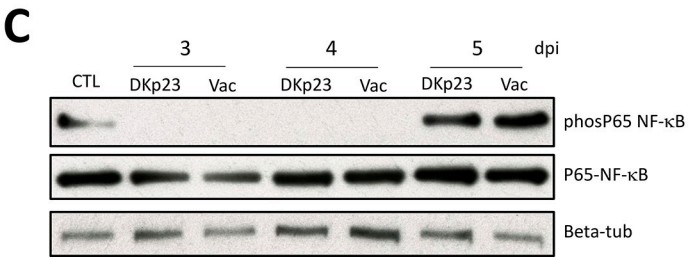

**Fig 4. Regulation of NF-κB nuclear translocation in the nucleus of Dkp23- or Vaccine-infected PK-15 cells. A** Immunofluorescence detection of p65NF-κB and bacteria in Dkp23- (middle panels), Vaccine-infected (bottom panels) and uninfected (upper panels) PK-15 cells at 2 and 8 hpi and 1, 3 and 5 dpi. Alexa-647-conjugated secondary antibody (red) and Alexa-488-conjugated secondary antibody (green) were used to detect p65 NF-κB (p65) and the bacteria (green), respectively. Nuclei were counterstained with DAPI (blue). Scale bars: 10 μm. **B** Ratio of p65 NF-κB nuclear staining intensity of Dkp23 (orange) and Enterisol (grey) infected PK-15 cells at 2 and 8 hpi and 1, 3 and 5 dpi to that of uninfected controls (blue). Mean values ± standard deviations are shown. Y-axis represent the ratio of p65 NF-κB nuclear staining intensity of infected PK-15 cells to that of uninfected controls. **C** Representative western blotting images of phosp65 NF-κB, p65 NF-κB and loading control Beta-tubullin. CTL, untreated cells; Dkp23, infection with Dkp23 strain; Vac, infection with the vaccine strain. hpi, hours post infection and dpi, days post infection.

was observed in Poly (I:C) treated cells as early as 2 hpt up to 48 hpt which was also associated with the concomitant increase of *IL-8, IL-6, TNF-α* transcript levels (S4–S6 Figs).

### A pulse of TNF-alpha halts the growth of *L. intracellularis in-vitro*

We then tested if activated NF-κB may alter *L. intracellularis* cytosolic growth. We exposed uninfected and *L. intracellularis* infected cells to TNF-alpha (n = 3), a known inducer of NF-κB signalling, for 24 hours and measured bacterial load at 4 dpi (n = 3) as depicted in the experimental plan in Fig 5A. Fig 5B shows that the *L. intracellularis* bacterial load was significantly reduced upon TNF-α treatment when PK-15 cells were challenged with either Dkp23 (*P<0.01*) or the vaccine strain (*P<0.05*).

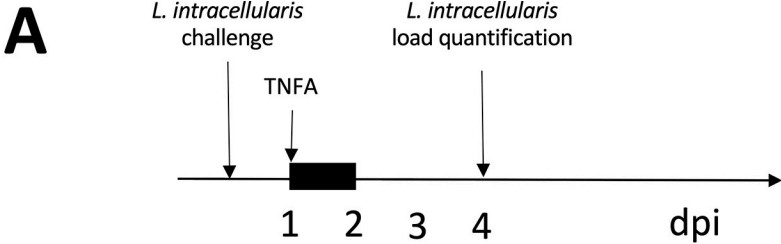

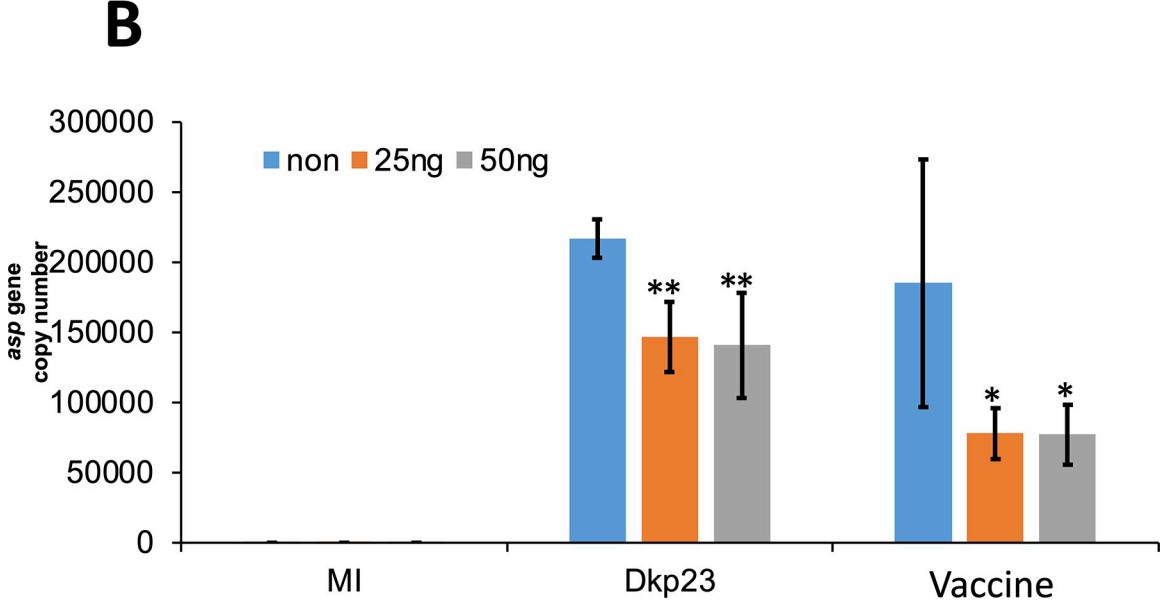

**Fig 5. A pulse of TNF-alpha halts the growth of *L. intracellularis* in-vitro. A** Schematic representation of the experiment. **B** PK-15 cells were seeded at 30% confluency in 24-wells culture dishes without penicillin/streptomycin supplement and were co-cultured with *L. intracellularis* (Control, mock infection (MI); DKp23; Vaccine; n = 3) at a MOI of 1 for 1 day. Wells were treated with TNF-α (blue, no TNF-α (non); orange, 25 ng of TNF-α; grey, 50 ng of TNF-α) for 24 h. Then wells were washed with PBS and culture medium. Finally, *L. intracellularis* bacteria was quantified by qPCR as described in the section of materials and methods on DNA preparation and qPCR analysis of *L. intracellularis* genomic copy number.

## Discussion

NF-κB signalling pathway plays a central role in innate immunity by regulating various cellular responses including apoptosis, cell proliferation, pro-inflammatory cytokines and chemokines expressions to restrict and aid in clearance of an infection. Results from both analysis of nuclear translocation of p65 NF-κB and of NF-κB–regulated downstream genes expressions suggest that the signalling pathway was significantly induced in both vaccine and Dkp23 infected cells, only at 5 dpi (Figs 3 and 4). This could be due to the significant increase in cytosolic bacterial load at 5 dpi compared to other earlier time points thus triggering the NF-κB signalling pathway (Fig 1). Activation of NF-κB signalling is a common feature observed during various obligate intracellular bacterial infection such as *Coxiella burnetti*, *Chlamydia spp*, *Rickettsia spp*, suggesting the presence as well as the growth of these obligate intracellular bacteria within its respective host cells would trigger the activation of NF-κB signalling [38–41]. However, NF-κB signalling is activated at early time points of the aforementioned infections (2–24 hpi), as compared to 5 dpi in the case of *L. intracellularis* infection. This timing difference could be due to the slower growth of *L. intracellularis*, differences between the infection models, as *L. intracellularis* infection requires actively dividing cells but not confluent culture, and differences between the pathogenicity and virulence factors of the bacteria. A significant decrease in the viability of both Dkp23 and vaccine infected cells were observed at 5 dpi, which could be due to the pro-apoptotic effect exhibited by induced expression of TNF-α, IFN-γ and other pro-apoptotic target genes of NF-κB signalling (Figs 3A, 3B and S2). Although *TNF-α* mRNA level was not significantly induced in *L. intracellularis* infected cells as compared to uninfected controls at 5 dpi, it is possible that a transient induction of *TNF-α* mRNA was missed in our investigation. The kinetics of NF-κB target genes expression (*TNF-α*, *IFN-γ*, *IL-6* and *IL-8*) appeared to vary as shown in RT-qPCR analysis in Poly (I:C) treated cells, consistent with previous observations (S4 Fig) [35,36]. Thus, a greater frequency in data collection in future would provide useful information regarding the overall kinetics of NF-κB target genes expressions. Moreover, a thorough examination of caspases activity and cleaved-caspases level would be of interest to investigate the pro-apoptotic activity observed 5 dpi, as mentioned earlier, NF-κB signalling activation in certain contexts may promote anti-apoptotic effects although this appeared to be less likely during acute infections in epithelial cells [42,43].

Immunofluorescence and RT-qPCR results from this study suggested that NF-κB signalling is either not induced or downregulated in *L. intracellularis*-infected cells at time points earlier than 5 dpi. Various intracellular pathogens especially viruses were known to inhibit NF-κB signalling to promote its reproduction and the survival of host cells [20,21]. *Chlamydia trachomatis*, an obligate intracellular bacterium is known to express a tail-specific protease (CT441) which targets and cleaves p65 NF-κB thus inhibits NF-κB signalling in infected HeLA cells [41]. Deubiquitination of inhibitor of κB (IκB) is observed during various intracellular bacterial infection on epithelial cells such as *Shigella flexineri*, *S. typhimurium*, *C. trachomatis*, Porcine reproductive and respiratory syndrome virus and Varicella-Zoster Virus, leading to the sequestration of p65 in the cytoplasm hence inhibits NF-κB signalling [44–48]. Conversely, *Rickettsia rickettsia*, another obligate intracellular bacterium is known to induce NF-κB signalling to promote the survival of infected blood endothelial cells [38]. The dual effects of NF-κB responses might be due to differences in the cell types used for infection, as the examples described previously including *L. intracellularis* were based in epithelial cells or fibroblast as opposed to *R. rickettsia* which specifically infects endothelial cells. Overall, intracellular pathogens may have evolved various mechanisms to interfere with steps of NF-κB signalling to promote the host cells survival and suppress pro-inflammatory responses in favour of the pathogen growth within the host cells. Further examination of different Rel and IκB family

proteins levels, its intracellular distribution and post-translational modifications during *L. intracellularis* infection could provide insights into whether NF-κB response is compromised during infection. Besides that, the use of dead *L. intracellularis* bacteria and bacterial lysate as controls is required to verify whether the bacterial growth or bacterial-derived factors are required for such mechanisms involved in delaying or inactivating NF-κB responses. Study by Oh et al. [22] reported the induced expression of several interferon responsive genes was observed in PHE *L. intracellularis* infected McCoy mouse fibroblast cells 48 hpi, suggesting an upregulation of interferon-dependant signalling during the early stages of infection. However, results from our study showed that *IFN-γ* mRNA levels of both vaccine and Dkp23 infected PK-15 cells remained similar to that of uninfected controls at 2 and 8 hpi and 1 and 3 dpi (Fig 3A). This discrepancy could be due to host cell sensitivity in response to *L. intracellularis* infection, but not the differences in the bacterial strain variants, because recent genomic phylogenetic analysis of multiple *L. intracellularis* strains indicates that DkP23 is genotypically close to PHE [32].

Our results show similar changes in NF-κB signalling target genes expression and p65 NF-κB nuclear translocation between Dkp23 and vaccine infected PK-15 cells at all-time points examined, suggesting the lack of significant differences in the pathogenicity between a live-attenuated and pathogenic strains during *in vitro* infection of PK-15 cells. This is consistent with previous studies showing the lack of overt cytopathic effects and no noticeable differences in the disease pathogenicity of *in vitro* infections using isolates from pigs with PIA and PHE, despite the significant differences in the clinical manifestation of both diseases *in vivo* [49–51]. The limited differences observed for NF-κB signalling target genes expression between Dkp23 and vaccine infected PK-15 cells in our study could be attributed to variation in the kinetics of infection progression *in vitro*, due to the vaccine having a significantly greater growth rate compare to strain Dkp23 (Fig 1A).

In summary, the data presented in this paper show 2 consistent experimental informations during *L. intracellularis* infection, 5 dpi, regardless of the strain pathogenicity: 1, an induction of IL-8 and 2, a strong nuclear localisation (Figs 4A, 4B and S5) and phosphorylation of p65 NF-κB (Fig 4C). The results from this study suggest possibility a compromised NF-κB responses during the early stages of *L. intracellularis* infection of PK-15 cells. We further showed that a pulse of TNF-α, an activator of the NF-κB signalling, reduced the growth of the pathogen in PK-15 cells. Taken together, these observations provide novel insights into the disease pathogenesis of *L. intracellularis* and indicate a potential strategy by which the intracellular bacteria may subvert the pathway leading to inflammatory response during intracellular proliferation and further echo our previous observation indicating that *L.intracellularis* may alter autophagy in infected ileum [12]. Interestingly, these observations are reminiscent of recent sophisticated mechanisms elaborated by invading microbes such as *M. tuberculosis* to promote their own survival and growth in macrophages [52]. Globally, our observations provide new knowledge and the impetus for additional studies to elucidate interactions between *L. intracellularis* and the host cell. However, it is important to note that the differences observed between *L. intracellularis* infection of PK-15 cells suggest that such conventional cell lines do not fully reproduce the disease pathogenesis of PE. Thus, an *in vitro* infection model which recapitulate features of the intestinal epithelium structure such as intestinal organoids would be a more appropriate model to studying *L. intracellularis* infection [53,54].

## Supporting information

**S1 Table. Primers used for PCR and qRTPCR.**
(TIF)

**S1 Fig. Magnification x2 of Fig 1B.**
(TIF)

**S2 Fig. Viability of *L. intracellularis.*** Percentage viability of Dkp23 (orange) and Vaccine (grey) infected PK-15 cells as compared to uninfected controls at 1, 3 and 5 dpi. Mean values ± standard deviations are shown. Y-axis represents percentage cell viability.
(TIF)

**S3 Fig. Regulation of IL8 cytokine in *L. intracellularis* infected PK-15 cells.** IL8 cytokine was measured using porcine cytokine antibody array A at 1, 3 and 5 dpi in Dkp23 or vaccine strains infected cells and in uninfected cells are shown. Y-axi s mean values in arbitrary unit ± standard deviations (n = 3).
(TIF)

**S4 Fig. Regulation of NF-κB target genes.** Fold change of NF-κB signalling target genes in Poly (I:C) treated (orange) over untreated PK-15 cells (blue) at 2, 8, 24 and 48 hpt. Mean values ± standard deviations are shown.*denotes $p<0.05$,**denotes $p<0.005$,****denotes $p<0.00005$.
(TIF)

**S5 Fig. Magnification X2 of Fig 4A.**
(TIF)

**S6 Fig. Immunofluorescence staining of p65 NF-κB in Poly (I:C) treated and untreated PK-15 cells at 2, 8, 24 and 48 hours post treatment.** Staining of p65 NF-κB (p65) was detected using Alexa-647 conjugated anti-rabbit antibody (red). Nuclei were counterstained with DAPI (blue). Scale Bar: 10 μm.
(TIF)

**S1 Raw images. Western blots used for Fig 4C.**
(TIF)

## Acknowledgments

We thank the clinical laboratory of the Royal (Dick) School of Veterinary studies in Edinburgh for the technical support during immune-detection of *L. intracellularis.*

## Author Contributions

**Conceptualization:** Huan W. Yang, Tahar Ait-Ali.

**Data curation:** Huan W. Yang, Tahar Ait-Ali.

**Formal analysis:** Huan W. Yang, Tahar Ait-Ali.

**Funding acquisition:** Tahar Ait-Ali.

**Investigation:** Huan W. Yang, Tuanjun Hu.

**Methodology:** Huan W. Yang, Tuanjun Hu.

**Resources:** Tuanjun Hu, Tahar Ait-Ali.

**Supervision:** Tahar Ait-Ali.

**Validation:** Tahar Ait-Ali.

**Writing – original draft:** Huan W. Yang, Tahar Ait-Ali.

**Writing – review & editing:** Huan W. Yang, Tuanjun Hu, Tahar Ait-Ali.

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
