## [Decision Letter · Decision Letter 0]

16 Apr 2024

PONE-D-24-01079Lawsonia intracellularis regulates Nuclear Factor-kB signalling pathway during infectionPLOS ONE

Dear Dr. AIT ALI,

Thank you for submitting your manuscript to PLOS ONE. After careful consideration, we feel that it has merit but does not fully meet PLOS ONE’s publication criteria as it currently stands. Therefore, we invite you to submit a revised version of the manuscript that addresses the points raised during the review process.

We look forward to receiving your revised manuscript.

Kind regards,

Md Bashir Uddin, DVM, PhD

Academic Editor

PLOS ONE

Journal Requirements:

Reviewers' comments:

Reviewer's Responses to Questions

**Comments to the Author**

1. Is the manuscript technically sound, and do the data support the conclusions?

Reviewer #1: Yes

Reviewer #2: Yes

2. Has the statistical analysis been performed appropriately and rigorously? 

Reviewer #1: Yes

Reviewer #2: Yes

3. Have the authors made all data underlying the findings in their manuscript fully available?

Reviewer #1: Yes

Reviewer #2: Yes

4. Is the manuscript presented in an intelligible fashion and written in standard English?

Reviewer #1: No

Reviewer #2: Yes

5. Review Comments to the Author

Reviewer #1: Lawsonia intracellularis is the causal agent of porcine proliferative enteropathy. However, current research on the interaction of L. intracellularis with the host is largely hindered by its fastidious, microaerophilic, and obligate intracellular nature. The manuscript PONE-D-24-01079 titled " Lawsonia intracellularis regulates Nuclear Factor-kB signaling pathway during infection" demonstrated that NF-Kb signaling level is induced when L. intracellularis bacterial load peaks at day 5 PI. Overall, the data presented here indicate a correlation between the induction of NF-kB signaling and the L.intracellularis bacterial load in PK-15 cell line. This research is interesting. However, some concerns should be placed before acceptance.

Major comments:

1. Line 112-113: “Cells were then co-cultured with L. intracellularis at a multiplicity of infection (MOI) of 1” , while in line 229-231, the method used to quantify L. intracellularis is qPCR, so how to determine the MOI is 1.

2. The quality of Figures was very poor, please provide the higher quality Figures again. In addition, please provide the magnified image in Figure 1B and 4A that shows the morphology of L. intracellular.

3. In the manuscript, the authors detected L. intracellular infection at 1, 3 and 5 dpi, but it takes 7 days to culture L. intracellular for a single generation, so whether authors detected the level of nuclear factor-kB signaling pathway at 7 days post-infection?

4. Were the results of the current study further validated in vivo?

5. Could authors describe the physiological significance of “NF-Kb signaling level is induced when L. intracellularis infection”

Minor comments:

1. Line 126-128: “As a positive control, 10µg/ml of Polyinosinic-polycytidylic acid (Poly (I:C), Invitrogen) was used to treat PK-15 cells for activating NF-�B [35, 36], the treated cell werecultured for 48 hours” might be “As a positive control, 10 µg/mL……were cultured for 48 hours”

2. It is suggested that the manuscript might be polished by a native English speaker.

Reviewer #2: The authors investigated the nuclear factor-κB (NF-κB)- regulated immune response against infection of a clinical strain Dkp23 and a live-attenuated Enterisol vaccine strain in PK-15 cells. They found that expression of NF-κB target genes TNF-α, IFN-γ, IL-6 and IL-8 were modulated during the course of infection. They also found that there was a significant increase in p65 NF-κB activation, including protein nuclear translocation and phosphorylation in L. intracellularis infected cells at 5 dpi. The methods were described clearly. However, I still have some comments.

Major comments:

1. In Figure 4C, authors detected the expression of phos-p65, p65 and β-tubulin in untreated and L. intracellularis infection cells. The basal expression of p-p65 was detected in the untreated cells, but completely undetectable in the L. intracellularis infected cells, at 3 and 4 dpi. Why the expression levels of p-p65 were not detected at 2, 8 hpi, and 1 dpi, as in the previous experiments? Is there a possibility that NF-κB signaling pathway was inhibited to facilitate the bacterium replication in the cytoplasm during the early phase of L. intracellularis infection, while NF-κB signaling pathway was activated to induce an inflammatory response, which aiding the bacterium escape the cell during the late phase infection. Additionally, besides detecting p-p65, can the levels of p65 in the cytoplasm and nucleus be simultaneously detected to demonstrate NF-κB activation?

2. In Figure 5B, authors shows that the L. intracellularis bacterial load was significantly reduced upon TNF-α treatment when PK-15 cells were challenged with either Dkp23 or the vaccine strain. TNF-α was added 1 d after L. intracellularis infection and the bacterial colonization was detected after 4 d of infection. During early infection, L. intracellularis likely needs to inhibit the host's inflammatory response to aid its proliferation in host cells. Therefore, the addition of TNF-α, activating the NF-κB signaling pathway, inhibits its colonization in the cells. This is consistent with the results in Figure 4C and can be further discussed.

Minor comments:

1. It is recommended to reduce the number of keywords.

2. Line 22, I believe 'PI' should be written as 'dpi'. Please, check this.

3. In some places throughout the text, a space is missing between the end of the subordinate clause and the principal clause. Please, check this. (incl. lines 23, 78, 213, 288).

4. Throughout the text, the time unit is sometimes the full name, and sometimes the abbreviation Please, make sure this is consistent throughout.

5. Line 25, I believe "PK-15" should be " PK-15 cells". Please check it.

6. Line 104-105: I believe "100 units/100 μg per mL " should be "100 units/ml". Please check it.

7. There are no spaces between numbers and units in the text. Please, check this. (incl. lines 120, 126, 127, 130, 139, 150, 159, 168, 194, 196, 200, 253, 255, 301, 312)

8. Please list the reference (incl. Line 128, 131)

9. I believe "days PI" should be "dpi". Please check it. (incl. line 151, 179, 188, 204, 254, 299, 303, 417,)

10. Line 269, I believe " DPI " should be "dpi". Please check it.

11. I believe " P65 " should be "p65". Please check it. (incl. line 297, 302, 306)

12. Line 314, I believe "P cell " should be "PK-15 cell". Please check it.

13. Line 322: I believe "qRT-PCR " should be "qPCR". Please check it.

14. I believe " DKp23 " should be "Dkp23". Please check it. (incl. line 307, 315)

15. I believe " TNFα " should be "TNF-α". Please check it. (incl. line 16, 259, 349)

16. Line 272: I believe "PK15 " should be "PK-15". Please check it.

17. I believe " IFN-ϒ" should be "IFN-γ". Please check it. (incl. line 262, 264)

18. Line 423, I believe " NF-KB " should be "NF-κB". Please check it.

19. Line 428, I believe " NF-kB " should be "NF-κB". Please check it.

6. PLOS authors have the option to publish the peer review history of their article (what does this mean?). If published, this will include your full peer review and any attached files.

Reviewer #1: No

Reviewer #2: No

---

## [Author Response · Author response to Decision Letter 0]

24 May 2024

Response to reviewer,

Reviewer comment:

The authors investigated the nuclear factor-κB (NF-κB)- regulated immune response against infection of a clinical strain Dkp23 and a live-attenuated Enterisol vaccine strain in PK-15 cells. They found that expression of NF-κB target genes TNF-α, IFN-γ, IL-6 and IL-8 were modulated during the course of infection. They also found that there was a significant increase in p65 NF-κB activation, including protein nuclear translocation and phosphorylation in L. intracellularis infected cells at 5 dpi. The methods were described clearly. However, I still have some comments.

Major comments:

1. In Figure 4C, authors detected the expression of phos-p65, p65 and β-tubulin in untreated and L. intracellularis infection cells. The basal expression of p-p65 was detected in the untreated cells, but completely undetectable in the L. intracellularis infected cells, at 3 and 4 dpi. 

a. Why the expression levels of p-p65 were not detected at 2, 8 hpi, and 1 dpi, as in the previous experiments? Is there a possibility that NF-κB signaling pathway was inhibited to facilitate the bacterium replication in the cytoplasm during the early phase of L. intracellularis infection, while NF-κB signaling pathway was activated to induce an inflammatory response, which aiding the bacterium escape the cell during the late phase infection. 

RESPONSE from authors:

The assessment of NF-κB signaling in L.intracellularis infected cells using western blot was an attempt to exploit well-established molecular tools. Based on staining of p65, our result indicated a significant increase in nuclear localization of p65 at 5 dpi but not at earlier time points post infection (Figure 4A), which correlated with the peak of bacterial load in PK-15 cells (Figure 1). Therefore, we decided to detect p-65 between 3 to 5 dpi via Western Blot to verify if the NF-κB signaling is activated at the peak of L. intracellularis bacterial load. Both the western blot data of p-65 (Fig 4C) and the observation of nuclear staining of p65 (Fig 4A) consistently indicated a correlation between L. intracellularis bacterial load and the activation of NF-κB. The background activation of p65 in uninfected control, as indicated by the presence of p-p65 (Fig 4C) as well as nuclear staining of p65 (Fig 4A) might be due to prolonged culture of pK-15 cells at high confluency. Nevertheless, the nuclear staining of p65 in L. intracellularis infected PK-15 cells was still significantly induced, as compared to uninfected samples at 5 dpi. We agree with the reviewer that assessing p-p65 at earlier time points would have been informative to increase our understanding of the early regulation of NF-κB signaling during L. intracellularis infection. Interestingly, the scrutiny of confocal images (Fig 4A) at 2 hpi, 8 hpi and 1 dpi indicates that there are no visible signal indicating the nuclear localization of p65 in infected cells harboring only few bacteria in the cytosol. This observation, however, is not quantitative and doesn’t constitute a proof of whether nuclear translocation of p65 and/or its phosphorylation status might be inhibited by the bacteria. The possibility of culturing L.intracellularis in-vitro as well as genetic modification of the bacteria, although technically demanding, will allow further investigations on early stage of L. intracellularis infection and its immune antagonistic mechanism(s), if any. 

b. Additionally, besides detecting p-p65, can the levels of p65 in the cytoplasm and nucleus be simultaneously detected to demonstrate NF-κB activation?

RESPONSE from authors:

We have evaluated p65 staining using confocal microscopy during infection as shown in Fig4A, which showed both nuclear and cytoplasmic distribution of p65. The current literature supports the notion of p65 nuclear translocation and its subsequent phosphorylation as the canonical signal activation pathway (J Inflamm Res. 2018; 11: 407–419). Furthermore, the ratio of infected to uninfected nuclear p65-NF-κB staining was quantified at all time points post infection in Fig 4B. We have also provided RT-qPCR results indicating induced level of cytokines such as IL-6 and IL-8 (Fig 3), in which NF-κB signaling was shown to upregulate the expression level of these cytokines in epithelial and endothelial cells (Eliott et al., 2001;Braiser, 2010)

2. In Figure 5B, authors shows that the L. intracellularis bacterial load was significantly reduced upon TNF-α treatment when PK-15 cells were challenged with either Dkp23 or the vaccine strain. TNF-α was added 1 d after L. intracellularis infection and the bacterial colonization was detected after 4 d of infection. During early infection, L. intracellularis likely needs to inhibit the host's inflammatory response to aid its proliferation in host cells. Therefore, the addition of TNF-α, activating the NF-κB signaling pathway, inhibits its colonization in the cells. This is consistent with the results in Figure 4C and can be further discussed.

RESPONSE to comment:

We thank the reviewer for the suggestion. We have now integrated in the “discussion section”, in the last paragraph, elements of discussion of the importance of our observation regarding the results of Fig 5. We have tried to minimize speculation regarding the possible immune antagonistic mechanism during early stages of L. intracellularis infection. Culturing L.intracellularis istechnically challenging and progress towards increasing our knowledge about bacterial-derived factors and the interactions with its host will be a long process. , 

Minor comments:

We have taken on board all the minor comments that the reviewer has made. 

1. It is recommended to reduce the number of keywords.

2. Line 22, I believe 'PI' should be written as 'dpi'. Please, check this.

3. In some places throughout the text, a space is missing between the end of the subordinate clause and the principal clause. Please, check this. (incl. lines 23, 78, 213, 288).

4. Throughout the text, some time units have full names, and some don't. Please, make sure this is consistent throughout.

5. Line 25, I believe "PK-15" should be " PK-15 cells". Please check it.

6. Line 104-105: I believe "100 units/100 μg per mL " should be "100 units ml−1". Please check it.

7. There are no spaces between numbers and units in the text. Please, check this. (incl. lines 120, 126, 127, 130, 139, 150, 159, 168, 194, 196, 200, 253, 255, 301, 312)

8. Please list the reference (incl. Line 128, 131)

9. I believe "days PI" should be "dpi". Please check it. (incl. line 151, 179, 188, 204, 254, 299, 303, 417,)

10. Line 269, I believe " DPI " should be "dpi". Please check it.

11. I believe " P65 " should be "p65". Please check it. (incl. line 297, 302, 306)

12. Line 314, I believe "P cell " should be "PK-15 cell". Please check it.

13. Line 322: I believe "qRT-PCR " should be "qPCR". Please check it.

14. I believe " DKp23 " should be "Dkp23". Please check it. (incl. line 307, 315)

15. I believe " TNFα " should be "TNF-α". Please check it. (incl. line 16, 259, 349)

16. Line 272: I believe "PK15 " should be "PK-15". Please check it.

17. I believe " IFN-ϒ" should be "IFN-γ". Please check it. (incl. line 262, 264)

18. Line 423, I believe " NF-KB " should be "NF-κB". Please check it.

19. Line 428, I believe " NF-kB " should be "NF-κB". Please check it.

References

1) Brasier, A.R. (2010) ‘The nuclear factor- B-interleukin-6 signalling pathway mediating vascular inflammation’, Cardiovascular Research, 86(2), pp. 211–218. doi:10.1093/cvr/cvq076. 

2) Elliott, C.L. “Nuclear Factor-Kappa B Is Essential for up-Regulation of Interleukin-8 Expression in Human Amnion and Cervical Epithelial Cells.” Molecular Human Reproduction, vol. 7, no. 8, 1 Aug. 2001, pp. 787–790, doi:10.1093/molehr/7.8.787. 

3) Giridharan, S. and Srinivasan, M. (2018) ‘Mechanisms of nf-κb P65 and strategies for therapeutic manipulation’, Journal of Inflammation Research, Volume 11, pp. 407–419. doi:10.2147/jir.s140188.

---

## [Editor Report · Decision Letter 1]

4 Jun 2024

PONE-D-24-01079R1Lawsonia intracellularis regulates Nuclear Factor-kB signalling pathway during infectionPLOS ONE

Dear Dr. AIT ALI,

Thank you for submitting your manuscript to PLOS ONE. After careful consideration, we feel that it has merit but does not fully meet PLOS ONE’s publication criteria as it currently stands. Therefore, we invite you to submit a revised version of the manuscript that addresses the points raised during the review process.

We look forward to receiving your revised manuscript.

Kind regards,

Md Bashir Uddin, PhD

Academic Editor

PLOS ONE

Journal Requirements:

Additional Editor Comments:

Author should specify the Reviewer responses (Reviewer 1 and reviewer 2) individually

---

## [Author Response · Author response to Decision Letter 1]

17 Jul 2024

The docs requested have been included in the submission

---

## [Editor Report · Decision Letter 2]

31 Jul 2024

PONE-D-24-01079R2Lawsonia intracellularis regulates Nuclear Factor-kB signalling pathway during infectionPLOS ONE

Dear Dr. ALI,

Thank you for submitting your manuscript to PLOS ONE. After careful consideration, we feel that it has merit but does not fully meet PLOS ONE’s publication criteria as it currently stands. Therefore, we invite you to submit a revised version of the manuscript that addresses the points raised during the review process.

We look forward to receiving your revised manuscript.

Kind regards,

Md Bashir Uddin, PhD

Academic Editor

PLOS ONE

**Additional Editor Comments:**

Author should response the following queries from reviewers:

Lawsonia intracellularis is the causal agent of porcine proliferative enteropathy. However, current research on the interaction of L. intracellularis with the host is largely hindered by its fastidious, microaerophilic, and obligate intracellular nature. The manuscript PONE-D-24-01079 titled " Lawsonia intracellularis regulates Nuclear Factor-kB signaling pathway during infection" demonstrated that NF-Kb signaling level is induced when L. intracellularis bacterial load peaks at day 5 PI. Overall, the data presented here indicate a correlation between the induction of NF-kB signaling and the L.intracellularis bacterial load in PK-15 cell line. This research is interesting. However, some concerns should be placed before acceptance.

Major comments:

1. Line 112-113: “Cells were then co-cultured with L. intracellularis at a multiplicity of infection (MOI) of 1” , while in line 229-231, the method used to quantify L. intracellularis is qPCR, so how to determine the MOI is 1.

2. The quality of Figures was very poor, please provide the higher quality Figures again. In addition, please provide the magnified image in Figure 1B and 4A that shows the morphology of L. intracellular.

3. In the manuscript, the authors detected L. intracellular infection at 1, 3 and 5 dpi, but it takes 7 days to culture L. intracellular for a single generation, so whether authors detected the level of nuclear factor-kB signaling pathway at 7 days post-infection?

4. Were the results of the current study further validated in vivo?

5. Could authors describe the physiological significance of “NF-Kb signaling level is induced when L. intracellularis infection”

Minor comments:

1. Line 126-128: “As a positive control, 10µg/ml of Polyinosinic-polycytidylic acid (Poly (I:C), Invitrogen) was used to treat PK-15 cells for activating NF-�B [35, 36], the treated cell werecultured for 48 hours” might be “As a positive control, 10 µg/mL……were cultured for 48 hours”

2. It is suggested that the manuscript might be polished by a native English speaker.

---

## [Author Response · Author response to Decision Letter 2]

23 Aug 2024

We have gathered the responses in a single document ( see below)

Response to reviewer 1

Reviewer 1 comment:

The authors investigated the nuclear factor-κB (NF-κB)- regulated immune response against infection of a clinical strain Dkp23 and a live-attenuated Enterisol vaccine strain in PK-15 cells. They found that expression of NF-κB target genes TNF-α, IFN-γ, IL-6 and IL-8 were modulated during the course of infection. They also found that there was a significant increase in p65 NF-κB activation, including protein nuclear translocation and phosphorylation in L. intracellularis infected cells at 5 dpi. The methods were described clearly. However, I still have some comments.

Major comments:

1. In Figure 4C, authors detected the expression of phos-p65, p65 and β-tubulin in untreated and L. intracellularis infection cells. The basal expression of p-p65 was detected in the untreated cells, but completely undetectable in the L. intracellularis infected cells, at 3 and 4 dpi. 

a. Why the expression levels of p-p65 were not detected at 2, 8 hpi, and 1 dpi, as in the previous experiments? Is there a possibility that NF-κB signaling pathway was inhibited to facilitate the bacterium replication in the cytoplasm during the early phase of L. intracellularis infection, while NF-κB signaling pathway was activated to induce an inflammatory response, which aiding the bacterium escape the cell during the late phase infection. 

RESPONSE from authors: The assessment of NF-κB signaling in L.intracellularis infected cells using western blot was an attempt to exploit well-established molecular tools. Based on staining of p65, our result indicated a significant increase in nuclear localization of p65 at 5 dpi but not at earlier time points post infection (Figure 4A), which correlated with the peak of bacterial load in PK-15 cells (Figure 1). Therefore, we decided to detect p-65 between 3 to 5 dpi via Western Blot to verify if the NF-κB signaling is activated at the peak of L. intracellularis bacterial load. Both the western blot data of p-65 (Fig 4C) and the observation of nuclear staining of p65 (Fig 4A) consistently indicated a correlation between L. intracellularis bacterial load and the activation of NF-κB. The background activation of p65 in uninfected control, as indicated by the presence of p-p65 (Fig 4C) as well as nuclear staining of p65 (Fig 4A) might be due to prolonged culture of pK-15 cells at high confluency. Nevertheless, the nuclear staining of p65 in L. intracellularis infected PK-15 cells was still significantly induced, as compared to uninfected samples at 5 dpi. We agree with the reviewer that assessing p-p65 at earlier time points would have been informative to increase our understanding of the early regulation of NF-κB signaling during L. intracellularis infection. Interestingly, the scrutiny of confocal images (Fig 4A) at 2 hpi, 8 hpi and 1 dpi indicates that there are no visible signal indicating the nuclear localization of p65 in infected cells harboring only few bacteria in the cytosol. This observation, however, is not quantitative and doesn’t constitute a proof of whether nuclear translocation of p65 and/or its phosphorylation status might be inhibited by the bacteria. The possibility of culturing L.intracellularis in-vitro as well as genetic modification of the bacteria, although technically demanding, will allow further investigations on early stage of L. intracellularis infection and its immune antagonistic mechanism(s), if any. 

b. Additionally, besides detecting p-p65, can the levels of p65 in the cytoplasm and nucleus be simultaneously detected to demonstrate NF-κB activation?

RESPONSE from authors: We have evaluated p65 staining using confocal microscopy during infection as shown in Fig4A, which showed both nuclear and cytoplasmic distribution of p65. The current literature supports the notion of p65 nuclear translocation and its subsequent phosphorylation as the canonical signal activation pathway (J Inflamm Res. 2018; 11: 407–419). Furthermore, the ratio of infected to uninfected nuclear p65-NF-κB staining was quantified at all time points post infection in Fig 4B. We have also provided RT-qPCR results indicating induced level of cytokines such as IL-6 and IL-8 (Fig 3), in which NF-κB signaling was shown to upregulate the expression level of these cytokines in epithelial and endothelial cells (Eliott et al., 2001;Braiser, 2010)

2. In Figure 5B, authors shows that the L. intracellularis bacterial load was significantly reduced upon TNF-α treatment when PK-15 cells were challenged with either Dkp23 or the vaccine strain. TNF-α was added 1 d after L. intracellularis infection and the bacterial colonization was detected after 4 d of infection. During early infection, L. intracellularis likely needs to inhibit the host's inflammatory response to aid its proliferation in host cells. Therefore, the addition of TNF-α, activating the NF-κB signaling pathway, inhibits its colonization in the cells. This is consistent with the results in Figure 4C and can be further discussed.

RESPONSE from authors : We thank the reviewer for the suggestion. We have now integrated in the “discussion section”, in the last paragraph, elements of discussion of the importance of our observation regarding the results of Fig 5. We have tried to minimize speculation regarding the possible immune antagonistic mechanism during early stages of L. intracellularis infection. Culturing L.intracellularis istechnically challenging and progress towards increasing our knowledge about bacterial-derived factors and the interactions with its host will be a long process. 

Minor comments:

We have taken on board all the minor comments that the reviewer has made. 

1. It is recommended to reduce the number of keywords.

2. Line 22, I believe 'PI' should be written as 'dpi'. Please, check this.

3. In some places throughout the text, a space is missing between the end of the subordinate clause and the principal clause. Please, check this. (incl. lines 23, 78, 213, 288).

4. Throughout the text, some time units have full names, and some don't. Please, make sure this is consistent throughout.

5. Line 25, I believe "PK-15" should be " PK-15 cells". Please check it.

6. Line 104-105: I believe "100 units/100 μg per mL " should be "100 units ml−1". Please check it.

7. There are no spaces between numbers and units in the text. Please, check this. (incl. lines 120, 126, 127, 130, 139, 150, 159, 168, 194, 196, 200, 253, 255, 301, 312)

8. Please list the reference (incl. Line 128, 131)

9. I believe "days PI" should be "dpi". Please check it. (incl. line 151, 179, 188, 204, 254, 299, 303, 417,)

10. Line 269, I believe " DPI " should be "dpi". Please check it.

11. I believe " P65 " should be "p65". Please check it. (incl. line 297, 302, 306)

12. Line 314, I believe "P cell " should be "PK-15 cell". Please check it.

13. Line 322: I believe "qRT-PCR " should be "qPCR". Please check it.

14. I believe " DKp23 " should be "Dkp23". Please check it. (incl. line 307, 315)

15. I believe " TNFα " should be "TNF-α". Please check it. (incl. line 16, 259, 349)

16. Line 272: I believe "PK15 " should be "PK-15". Please check it.

17. I believe " IFN-ϒ" should be "IFN-γ". Please check it. (incl. line 262, 264)

18. Line 423, I believe " NF-KB " should be "NF-κB". Please check it.

19. Line 428, I believe " NF-kB " should be "NF-κB". Please check it.

References

1) Brasier, A.R. (2010) ‘The nuclear factor- B-interleukin-6 signalling pathway mediating vascular inflammation’, Cardiovascular Research, 86(2), pp. 211–218. doi:10.1093/cvr/cvq076. 

2) Elliott, C.L. “Nuclear Factor-Kappa B Is Essential for up-Regulation of Interleukin-8 Expression in Human Amnion and Cervical Epithelial Cells.” Molecular Human Reproduction, vol. 7, no. 8, 1 Aug. 2001, pp. 787–790, doi:10.1093/molehr/7.8.787. 

3) Giridharan, S. and Srinivasan, M. (2018) ‘Mechanisms of nf-κb P65 and strategies for therapeutic manipulation’, Journal of Inflammation Research, Volume 11, pp. 407–419. doi:10.2147/jir.s140188. 

Response to reviewer 2:

Major comments

1. Line 112-113: “Cells were then co-cultured with L. intracellularis at a multiplicity of infection (MOI) of 1” , while in line 229-231, the method used to quantify L. intracellularis is qPCR, so how to determine the MOI is 1. How did you measure the MOI?

Response from author : L. Intracellularis infections were carried out on PK-15 cells at 30% confluency in 24 wells-tissue culture dish. We estimated the number of PK-15 cells in each well based on the surface area occupied by cells, via a standard of 0.24*10^6 cells per 24 well (30% confluency would be approximately 0.08*10^6 cells per 24 well). We determined the L. intracellularis bacteria quantity in each stock produced, via qPCR in copy number of LI AspA gene. These allowed a determination of the MOI. Based on the confocal images shown in Fig 1 and Fig 4A indicate that our estimate of MOI of 1 early during infection were right.

2. The quality of Figures was very poor, please provide the higher quality Figures again. In addition, please provide the magnified image in Figure 1B and 4A that shows the morphology of L. intracellular. 

Response from author : We have now improved image quality with the PACE tool suggested by Plos and provide magnified images for Fig 1B and Fig 4A in supplemental data as S2 Fig and S6 Fig, respectively.

3. In the manuscript, the authors detected L. intracellular infection at 1, 3 and 5 dpi, but it takes 7 days to culture L. intracellular for a single generation, so whether authors detected the level of nuclear factor-kB signaling pathway at 7 days post-infection? We did not detect the level of NFKB at 7Days pi?

Response for authors: Culture of L. Intracellularis was terminated at 5 days dpi as we started to see host cell death beyond this time point of infection as well as for mock infected cells although less obvious and/or frequent, suggesting that time points post 5 days would not be informativefor L. intracellularis infection therefore we did not assay Nf-KB activity at 7 dpi.

4. Were the results of the current study further validated in vivo? 

Response from authors: The work was performed in vitro only. 

5. Could authors describe the physiological significance of “NF-Kb signaling level is induced when L. intracellularis infection”: We need to think about this one

Response from authors: NFKB transcription family plays an important role in innate immune response whereby it is activated by an array of cellular pattern recognition receptors, leading to the expression and/or signal transduction of cytokines and chemokines affecting multiple aspects of cellular processes. Our results indicated an association between activation of NFKB signalling and the growth of L. Intracellularis in PK-15 cells in vitro therefore indicating that the host cell innate immune response is triggered during active replication of the bacteria, which was accompanied by enhanced expression of several NFKB downstream effector genes (detected in this study). Although we did not look into the activation of NFKB signaling during L. Intracellularis infection in vivo, one could speculate the effects, if any, of NFKB signalling activation and the activity of its effector cytokines/chemokines on the disease pathogenesis of PE. One of the main histopathology of PE is the dilated intestinal crypts with neutrophil infiltration and we observed a significant increase in IL-8 expression during L. Intracellularis infection in vitro, which itself is a potent chemokine targeting neutrophils and granulocytes. 

Minor comments:

1. Line 126-128: “As a positive control, 10µg/ml of Polyinosinic-polycytidylic acid (Poly (I:C), Invitrogen) was used to treat PK-15 cells for activating NF-kB [35, 36], the treated cell werecultured for 48 hours” might be “As a positive control, 10 µg/mL……were cultured for 48 hours”

Response from authors: We have edited the text accordingly.

2. It is suggested that the manuscript might be polished by a native English speaker.

Response from authors: We have taken on board this suggestion

---

## [Decision Letter · Decision Letter 3]

4 Sep 2024

Lawsonia intracellularis regulates Nuclear Factor-kB signalling pathway during infection

PONE-D-24-01079R3

Dear Dr. ALI,

We’re pleased to inform you that your manuscript has been judged scientifically suitable for publication and will be formally accepted for publication once it meets all outstanding technical requirements.

Kind regards,

Md Bashir Uddin, PhD

Academic Editor

PLOS ONE

Additional Editor Comments (optional):

Reviewers' comments:

Reviewer's Responses to Questions

**Comments to the Author**

1. If the authors have adequately addressed your comments raised in a previous round of review and you feel that this manuscript is now acceptable for publication, you may indicate that here to bypass the “Comments to the Author” section, enter your conflict of interest statement in the “Confidential to Editor” section, and submit your "Accept" recommendation.

Reviewer #1: All comments have been addressed

2. Is the manuscript technically sound, and do the data support the conclusions?

Reviewer #1: Yes

3. Has the statistical analysis been performed appropriately and rigorously? 

Reviewer #1: Yes

4. Have the authors made all data underlying the findings in their manuscript fully available?

Reviewer #1: Yes

5. Is the manuscript presented in an intelligible fashion and written in standard English?

Reviewer #1: Yes

6. Review Comments to the Author

Reviewer #1: The authors have responded to all comments and made appropriate and detailed revisions to the manuscript.

7. PLOS authors have the option to publish the peer review history of their article (what does this mean?). If published, this will include your full peer review and any attached files.

Reviewer #1: No

---

## [Editor Report · Acceptance letter]

18 Sep 2024

PONE-D-24-01079R3 

PLOS ONE

Dear Dr. Ait-Ali, 

I'm pleased to inform you that your manuscript has been deemed suitable for publication in PLOS ONE. Congratulations! Your manuscript is now being handed over to our production team.

Kind regards, 

on behalf of

Dr. Md Bashir Uddin 

Academic Editor

PLOS ONE